# The efficacy of promoting sustained shared thinking through the use of activity books on parental empowerment; A quasi-experimental study

**Kamolvisa Techapoonpon**[1], **Wisarat Pruttithavorn**[1*], **May Sripatanaskul**[2], **Niyata Limpiti**[2], **Kahwei Yoong**[2]

**1** Department of Psychiatry, Faculty of Medicine Vajira Hospital, Navamindradhiraj University, Bangkok, Thailand, **2** LUKKID Co., Ltd., Bangkok, Thailand

* wisarat@nmu.ac.th (WP)

## Abstract

### Background

Parental empowerment serves as an essential driving force in promoting child development during the early childhood period. Sustained shared thinking is a method involving two or more individuals engaging in a shared cognitive process through sustained conversation. This principle has proven to be an effective intervention for intellectual stimulation in preschool-aged children. We hypothesize that supporting parents in practicing sustained shared thinking would foster parental empowerment by allowing them to play an active role in the positive change of their children. This study investigates the effectiveness of enhancing sustained shared thinking through self-directed activity books in promoting parental empowerment.

### Methods

This research was a quasi-experimental study conducted from January 2, 2023, to April 24, 2023. The participants (n = 55) were mothers or fathers of children aged four to six years. Each parent received sustained shared thinking-stimulating activity books on a weekly basis over a period of six weeks. Empowerment assessments were made at four points in time: pre-intervention, the third week of intervention, the sixth week of intervention, and three months after completing the intervention.

### Results

Sustained shared thinking-stimulating activity books significantly increased parental empowerment. The mean empowerment score increased from baseline by 30.79% (95% CI [24.66, 36.92]) after three weeks of activities and increased by 115.36% (95% CI [105.19, 125.54]) after six weeks of activities. Empowerment scores

**Data availability statement:** All relevant data are within the paper.

**Funding:** This research received funding from the research funds of the Faculty of Medicine, Navamindradhiraj University. (027/2566). The funders had no role in study design, data collection and analysis, decision to publish, or preparation of the manuscript.

**Competing interests:** The author MS, NL and KY are employee of LUKKID Co., Ltd., (Bangkok, Thailand). This does not alter our adherence to PLOS ONE policies on sharing data and materials.

remained relatively stable at 114.11% (95% CI [103.9, 124.32]) when assessed three months post-intervention.

## Conclusions

Stimulation of sustained shared thinking through the use of activity books over six weeks significantly increases parental empowerment. The effect of parental empowerment is observable from the third week after the activities and persists for up to three months after the completion of the intervention. The format of this intervention is primarily self-directed activity, providing a solution that can be replicated and further developed in the future.

**Clinical trial registration:** Thai Clinical Trials Registry (TCTR20221203002)

## Introduction

The early childhood period refers to the time from birth to the age of six [1]. Due to the rapid growth of the brain during this critical stage, it is of utmost importance to maximize the development of intellectual and emotional intelligence in humans. Once the window of opportunity for optimal development has passed, reversing the loss is difficult [1]. Therefore, developmental stimulation during this period is crucial for ensuring optimal growth, for which parents play a key role because children typically spend most of their time with their parents [2]. However, many young children often experience insufficient developmental stimulation within their homes, leading to a significant loss of developmental opportunities. Survey has shown that 43% of primary school children in low- to middle-income countries experience under-stimulation from their families [3]. This issue is compounded by parents allowing their children to spend excessive time on smartphones, which has contributed to a significant rise in smartphone addiction among young children [4]. One of the primary reasons for this neglect in developmental stimulation is found to be low parental empowerment [5].

Parental empowerment refers to the process or outcome within the psyche of parents, subsequently manifest in whether they are able to control, manage and effectively fulfill key parenting responsibilities, such as providing emotional support, setting appropriate boundaries, and engaging in their child's learning and development [6]. Empowering parents positively influences the development of their children. A previous study found high levels of parental development to be associated with better decision-making regarding parenting and management of family issues [5]. Other studies have found parental empowerment to be associated with better overall child's wellbeing and academic performance, improved physical and emotional development, reduced child stress and behavioral problems [5,7–9]. On the contrary, low parental empowerment is associated with increased family conflict and diminished motivation to assist their children in difficult situations [5].

For the reasons discussed, supporting parental empowerment during early childhood, the critical developmental stage is of paramount importance. Our team of researchers aims to empower parents of young children by designing a parental

empowerment intervention based on the principle of sustained shared thinking (SST) [10]. This principle involves two or more individuals engaging in a shared cognitive process, where thoughts are exchanged through conversation, with both parties making sustained contributions [10]. Studies have shown that SST is an effective method for intellectually stimulating preschool-aged children [11,12]. Based on these findings, the researchers hypothesize that SST-based interventions can significantly enhance parental empowerment, since the intervention encourages parents to take an active role rather than passive [13]. When parents perceive themselves as active agent of their children's growth, this sense of self-efficacy becomes a crucial component of empowerment [13,14].

Our goal is to create an SST-stimulating intervention and assess its effectiveness in enhancing parental empowerment. We also evaluated children's development in cognitive and literacy skills from parent's perspective to measure the growth from the parent-led intervention. According to previous studies, it was found that self-help activity books are an effective medium for interventions, successfully transferring knowledge and shifting perspectives in both adults and preschool-aged children [15,16]. Therefore, we developed self-help activity workbooks, featuring a vibrant appearance and clear, simple instructions that enable parents to independently carry out SST-stimulating activities with their children. This particular intervention is designed for parents and children aged 4–6 years. To our knowledge, no prior research has specifically investigated the effects of SST interventions on parental empowerment. The findings from this study will inform the sustainable development of family support systems in the future.

## Materials and methods

### Study design

This single-arm, nonrandomized, quasi-experimental study was conducted from January 2, 2023 to April 24, 2023 and approved by the Institutional Review Board of the Faculty of Medicine Vajira Hospital (COA 091/2565). The sample size calculation was performed using the G*Power program (v.3.1.9.7). A one-group design with two-tailed test was performed, with parental empowerment specified as the outcome variable. The analysis parameters included an effect size of 0.5 (medium), a significance level (α) of 0.05, and a power of 0.9. The required total sample size was determined to be 44 [17]. This was increased by 20% to account for potential attrition, resulting in a final sample of 55 parents [18].

### Participants criteria, recruitment, and consent

Inclusion Criteria: Participants included parents aged over 18, who were biological mothers or fathers, cohabiting with their 4–6-year-old child, and scoring below 60% on the parental empowerment questionnaire, which indicated a lack of parental empowerment [19]. Each participant must come from a separate household.

Exclusion Criteria: Participants were excluded if the parent had a mental health condition, if the child had a neurodevelopmental disorder, or if the child had a physical condition requiring regular medication or physician follow-up. Participants could voluntarily withdraw from the study at any time.

The parental empowerment research program was promoted through schools and community networks in Bangkok using fliers that included contact information. Interested and eligible parents contacted the researchers directly through chat application. The screening survey for eligibility was performed via Google Form, asking questions according to inclusion and exclusion criteria. After screening, eligible participants (n = 55) provided written informed consent online via Google Form, confirming their agreement by typing their names and checking the consent box. Participants received an incentive of 300 THB upon completing each workbook.

### Instruments

**Evaluation instruments.** Demographic questionnaire – This questionnaire included parent's sex, age, family status, number of children, education, occupation, type of house, income, time spent with the child per day, the child's age, sex, and difficulty of the child to parents (subjective feeling).

Empowerment questionnaire – The researchers employed this questionnaire to measure empowerment level of the parents. This tool was used to identify and recruit subjects with low parental empowerment for our study and follow up the result after participating in the intervention. The researchers developed this questionnaire in full by using the theory of Zimmerman [20] as a backbone. We formulated questions corresponding to each domain proposed by Zimmerman and conducted confirmatory factor analysis (CFA) to assess whether the questions aligned with their respective domains. Zimmerman categorizes psychological empowerment into 12 subheadings within 3 components, consisting of the intra-personal component (4 subheadings: item 1–4), the interactional component (5 subheadings: item 5–9), and lastly, the behavioral component (3 subheadings: item 10–12). This questionnaire is a self-rating instrument for all 12 subheadings using the 5-point Likert scale format (1: strongly disagree, 2: disagree, 3: uncertain, 4: agree and 5: strongly agree) (Table 1). Each item had content validity index (CVI) > 0.79 [21]. In performing CFA, we used maximum likelihood estimation (ML) method and the model's goodness of fit. The CFA model of parental empowerment demonstrated an acceptable fit to the empirical data. The model yielded a chi-square ($\chi^2$) value of 48.519 was non-significant (p = .333, df = 45) and a relative chi-square ($\chi^2/df$) of 1.078, which is below the threshold of 2, indicating a good model fit. Additionally, the Comparative Fit Index (CFI) and Tucker–Lewis Index (TLI) were 0.910 and 0.884, respectively, both exceeding the criterion of 0.97. The Standardized Root Mean Squared Residual (SRMR) and Root Mean Squared Error of Approximation (RMSEA) were 0.075 and 0.062, respectively, both below the accepted threshold of 0.08. Taken together, these indicate that the specified confirmatory factor model aligns well with the empirical data and is considered acceptable [22]. Additionally, the Cronbach's alpha coefficient for this questionnaire was calculated to be 0.76, based on a sample of 160 participants. In which, the sample size was determined based on Cronbach's alpha estimation formula, assuming an expected Cronbach's alpha of 0.8, a precision of 0.05, a confidence level of 95%, 12 items, and an anticipated dropout rate of 10% [23,24].

Questionnaire evaluating children's development in cognitive and literacy skills – This questionnaire was listed by our team. It consists of 10 items for parents to evaluate their child's cognitive and language improvement after receiving intervention. The responses are divided into three options: 0 = no improvement, 1 = slight improvement, and 2 = significant improvement. Cognitive development is assessed in six areas based on Bloom's taxonomy: remembering (ability to recall content, repeat, and reproduce), understanding (ability to explain understanding in one's own words), applying (ability to apply knowledge in practical situations knowledge to practice), analyzing (ability to link cause and effect), evaluating (ability to compare, judge, and conclude logically), and creating (ability to integrate existing knowledge to extend, produce, or generate something new) [25]. Literacy skills are assessed in four areas: listening, speaking, reading, and writing [26]. An

**Table 1. Empowerment questionnaire (English version).**

| Items | strongly disagree | disagree | uncertain | agree | strongly agree |
|---|---|---|---|---|---|
| 1. I believe that I have an influence on my child's growth and development. | | | | | |
| 2. I feel confident in my ability to help my child grow and develop. | | | | | |
| 3. I am motivated to help my child grow and develop. | | | | | |
| 4. I make an effort to learn new ways to help my child grow and develop. | | | | | |
| 5. I can find useful information and knowledge for my child's development. | | | | | |
| 6. I understand that there are external factors that affect my child's growth and development. | | | | | |
| 7. I can develop and use skills that best support and enhance my child's learning. | | | | | |
| 8. I can apply the acquired skills to promote other areas of my child's life. | | | | | |
| 9. I can effectively use my knowledge and resources to promote my child's development. | | | | | |
| 10. I have become more involved in my child's learning and development community. | | | | | |
| 11. I am able to assist other parents if they need help. | | | | | |
| 12. I can manage my emotions and personal life when under stress. | | | | | |

Exploratory Factor Analysis (EFA) with Varimax rotation was conducted to evaluate the reliability and construct validity of a questionnaire assessing cognitive development and literacy skills. EFA was chosen due to the preliminary nature of the instrument and the sample size (n = 55) [26], with Varimax rotation applied to simplify the interpretation of factor loadings by maximizing the variance of squared loadings across factors [27]. The Kaiser-Meyer-Olkin (KMO) measure of sampling adequacy was 0.72, and Bartlett's test of sphericity was significant ($\chi^2(45) = 123.56$, $p < 0.001$), confirming the suitability of the data for factor analysis. A two-factor solution was identified, explaining 68.2% of the variance, with item loadings ranging from 0.61 to 0.85 for cognitive development and 0.58 to 0.83 for literacy skills. Cronbach's alpha coefficients were 0.81 for the cognitive development subscale and 0.79 for the literacy skills subscale, indicating internal consistency and supporting the questionnaire's reliability and construct validity.

**Intervention instruments.** SST-stimulating activity books – The activity books series consist of 6 books; each book contains activities to be carried out over a period of one week. Each activity book contains a different theme. However, the general format is the same throughout the series. Each book has 3 sections. The first section explains the concept of sustained shared thinking, its rationale, principles and methods, examples of dialogue that parents could use with their children to stimulate thinking, including do's and don'ts. The second part consists of 5 activities under the same theme. The third final section is for reflection: consisting of 5 open questions for parents to answer after having carried out all 5 activities in the book by the end of each week. The reflection questions consist of 1). the feelings of the parents regarding parental empowerment after having completed the activities. 2). How their thoughts and attitudes towards parenting have changed 3). The usefulness of these activities for the parents 4). The usefulness of these activities for the children from the parent's perspective and 5). Any feedback and suggestions the parents might have. The full details of the activity books are listed in Table 2.

## Data collection

After passing eligibility criteria, all the participants then received demographic questionnaire via Google Form, after which they received a weekly activity book by mail over a six-week period. To verify completion and engagement, participants were required to submit photographs of themselves performing activities with their children through a direct messaging application, along with reflective responses provided via Google Form. The efficacy of the intervention in enhancing parental empowerment was assessed at four specific points in time: before the intervention (to determine eligibility), after three weeks, after six weeks, and three months following completion of the intervention. To measure change after receiving interventions, children's cognitive and literacy skills development were evaluated at three intervals: after three weeks, after six weeks, and three months post-intervention. The detailed participant flow, adhering to the Consolidated Standards of Reporting Trials (CONSORT) guidelines [28,29], is depicted in Fig 1.

## Data analysis

Descriptive statistics were used for demographic data. Variables were presented as means and standard deviation for continuous data and as frequency and percentages for categorical data. To assess the effects of the program on changes in the outcomes, analyses were conducted with the use of a linear mixed-effects model with an autoregressive correlation matrix adjusted for baseline value, employing a two-tailed test.

We used intent-to-treat (ITT) analysis with a last observation carried forward (LOCF) approach for missing data. Demographic analysis of dropout (n = 9) revealed significant differences between the dropout and complete groups in terms of education level (p = 0.002), marital status (p = 0.006), and number of children (p = 0.033), suggesting that certain participant characteristics may be associated with attrition. However, Little's MCAR (Missing Completely at Random) test yielded a test statistic of 0.0 with a p-value of 1.0, indicating that missingness was completely at random. This result supports the validity of using LOCF, as MCAR ensures that missing data do not introduce systematic bias, making LOCF an

**Table 2. Content of sustained shared thinking stimulating activity.**

| Structure | Section 1<br>Introduction | Section 2<br>Activities | Section3<br>Reflections |
|---|---|---|---|
| Week 1<br>Theme:<br>Tooth brushing | Rationale, principle and guidance for sustained shared thinking | Read the story "The bad-breathed dragon" and engage the child about the story (sample conversational cues provided)<br>Discuss with the child what a toothbrush for a dragon would look like (paper toothbrush cutout provided for coloring)<br>Discuss with the child about making toothpaste for the dragon by drawing and coloring the toothpaste on paper provided.<br>Discuss with the child about the different types of teeth (illustrations provided)<br>Discuss how to use dental floss and carry out an activity using an ice tray and playdough to represent teeth and food stuck in between the teeth and use rope to represent floss. | Five open-ended questions |
| Week 2<br>Theme:<br>Toileting | | Discuss with the child about toileting, then guide the child to apply stickers provided to learn the steps for latrine use.<br>Watch an animated video clip of a hippopotamus going to the toilet loudly, then discuss about the clip (sample conversational cues provided)<br>Discuss with the child about how to decorate the bathroom to make going to the toilet fun and less intimidating, then carry out an activity to cooperatively invent items to be used in the household toilet.<br>Carry out an activity to craft a toy from toilet rolls under the stimulation and encouragement from parents.<br>Discuss with the child about what items are needed when using a public toilet, encourage the child to pack a bag containing items for use in the toilet using existing items in the home. | |
| Week 3<br>Theme:<br>Emotions | | Read the story "The Grumpy Monkey" and engage the child about the story (sample conversational cues provided)<br>Discuss with the child to explore about emotions relating to anger, look at pictures of children with various emotions provided, then engage the child to explore different emotions (sample conversational cues provided)<br>Using pictures provided to discuss with the child about the management of emotions, with anger in particular. (sample conversational cues provided)<br>Using the wheel of emotion provided to discuss with the child about different emotions.<br>Engage in an activity to create a dance routine together to relieve anger, using any music preferred by the child. | |
| Week 4<br>Theme:<br>Fear and Courage | | Discuss with the child to explore the emotion of fear, share experiences of fear between the parent and child. Explore ideas of how to deal with fear.<br>Engage in an activity to transform the child into a superhero; design and create a costume and mask for a hero (a cutout mask is provided in the activity book). Then, practice how to move like a hero (there are 20 examples of movements provided, e.g., standing on one leg or running on the same spot for 10 seconds)<br>Watch an animated video clip "Tonkla is not scared anymore", discuss the child about the story (sample conversational cues provided)<br>Discuss with the child about daily life situations which are safe or unsafe for the child to help others. Examples are provided such as a friend who is drowning, or has fallen over, or is crying because they miss their parents. Ask the child how they would respond to each of these situations<br>Discuss with the child to explore how the child overcome his/her fear, then write this down in a superhero certificate of achievement, awarded to the child. | |

*(Continued)*

**Table 2.** (Continued)

| Structure | Section 1<br>Introduction | Section 2<br>Activities | Section3<br>Reflections |
|---|---|---|---|
| Week 5<br>Theme:<br>Family | | Engage in an activity with the child, looking for leaves near the house to color and decorate to represent people in the family, then ask the child to discuss which leaves represent who in the family (examples provided in the activity book).<br>Play a game with the child, taking turns drawing family members and asking the other person to guess who the drawing represents, in addition, try to ask thought provoking questions such as "how did you know who this was?", or "how else could we draw this person?" etc.<br>Discuss with the child about their relatives and the provinces they reside in (a map of Thailand is provided). Examples of conversational cues are provided, such as discussing their experience visiting relatives, or which other relatives they would like to visit in other provinces etc.<br>Engage in an activity with the child, write in a notebook provided in the activity book with template questions about family, pertaining details such as, "my mother's name is…., mother likes to eat…, my family likes to eat…… etc.<br>Watch an animated video clip about a family creating a box of happiness, then discuss with the child (sample conversational cues provided) | |
| Week 6<br>Theme:<br>Relationships and Creativity | | Read the story "Penguins want to fly" then discuss with the child about the story (sample conversational cues provided).<br>Discuss with the child about their friends such as "who is your closest friend?", "what do you have in common with your friend?" and "how are your friends different from you?" (sample questions provided)<br>Discuss with the child, imagine a scenario they could encounter with their friends, for example, "what would you do when you meet a new friend?", "what would you do if someone took something from you?" or, "what would you do if you want to play with a toy your friend is also playing with?".<br>Engage in an activity with the child, imagine and create a flying machine that can help penguins fly by using existing objects around the house.<br>Discuss with the child about things that some animals can do and cannot do. Engage in an activity with the child, using the provided material, look at a table containing different animals ranging from fish to lions; for each animal, ask the child what each animal can or cannot do. For example, "Can this animal run, jump or swim?". Lastly, discuss with the child what they can or cannot do. | |

appropriate method for imputing missing values in this study. Statistical analysis was performed with Stata 13.0 software (StataCorp, College Station, TX, USA). P-value < 0.05 indicates statistical significance.

## Results

### Characteristics of the participants

A total of 55 volunteers participated, with details outlined in Table 3. The participating parents had an average age of approximately 34.58 years (SD = 6.6), with 94.5% (n = 52) being mothers. About 38.2% (n = 21) had only one child, 50.9% (n = 28) of the parents lived together, and 23.6% (n = 13) spent less than 4 hours per day with their child, excluding sleep time. The children had an average age of 4.95 years, with 47.3% (n = 26) being male and 52.7% (n = 29) being female.

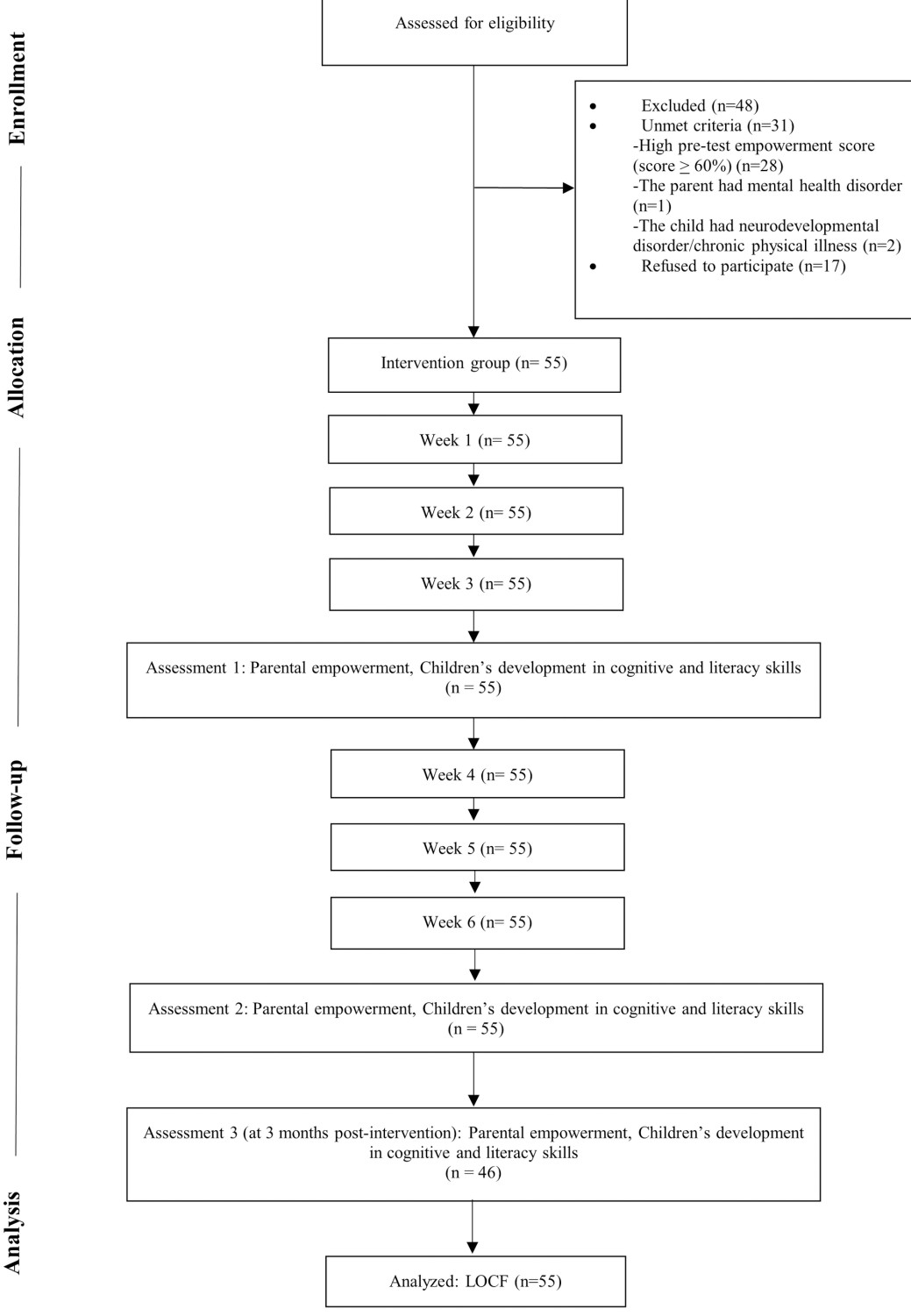

**Fig 1. Participant Flow.**

**Table 3. Characteristics of the participants.**

| Data | Mean (SD) | n | (%) |
|---|---|---|---|
| Parent's sex | | | |
| _Father_ | | 3 | (5.5) |
| _Mother_ | | 52 | (94.5) |
| Parent's age; year | 34.58 (6.60) | | |
| Family status | | | |
| _Spouses live together_ | | 28 | (50.9) |
| _Spouses live separately_ | | 27 | (49.1) |
| Number of children | | | |
| _1_ | | 21 | (38.2) |
| _2_ | | 25 | (45.5) |
| _More than 2_ | | 9 | (16.4) |
| Education | | | |
| _None_ | | 1 | (1.8) |
| _Primary school_ | | 4 | (7.3) |
| _High school_ | | 15 | (27.3) |
| _Diploma_ | | 9 | (16.4) |
| _Bachelor's degree_ | | 24 | (43.6) |
| _Master's degree or higher_ | | 2 | (3.6) |
| Occupation | | | |
| _Unemployed_ | | 2 | (3.6) |
| _Civil servant_ | | 8 | (14.5) |
| _Company employee_ | | 19 | (34.5) |
| _Freelancer_ | | 20 | (36.4) |
| _Business owner_ | | 6 | (10.9) |
| Income | | | |
| _0–5,000 THB_ | | 6 | (10.9) |
| _5,001–10,000 THB_ | | 13 | (23.6) |
| _10,001–15,000 THB_ | | 15 | (27.3) |
| _15,001–20,000 THB_ | | 7 | (12.7) |
| _20,001–25,000 THB_ | | 6 | (10.9) |
| _>25,000 THB_ | | 8 | (14.5) |
| Family type | | | |
| _Nuclear family_ | | 16 | (29.1) |
| _Extended family_ | | 39 | (70.9) |
| Type of house | | | |
| _Detached/Semi-detached house_ | | 11 | (20.0) |
| _Town house/Town home_ | | 12 | (21.8) |
| _Commercial building_ | | 2 | (3.6) |
| _Flat/Dormitory/Apartment_ | | 6 | (10.9) |
| _Condominium_ | | 1 | (1.8) |
| _Slum_ | | 23 | (41.8) |
| Time spent with the child (excluding sleep time) | | | |
| _Less than 2 hours per day_ | | 2 | (3.6) |

_(Continued)_

**Table 3.** (Continued)

| Data | Mean (SD) | n | (%) |
|---|---|---|---|
| 2 to less than 4 hours per day | | 11 | (20.0) |
| 4 to less than 6 hours per day | | 19 | (34.5) |
| 6 hours or more per day | | 23 | (41.8) |
| Child's age; year | 4.95 (0.80) | | |
| 4 | | 19 | (34.5) |
| 5 | | 20 | (36.4) |
| 6 | | 16 | (29.1) |
| Child's gender | | | |
| Male | | 26 | (47.3) |
| Female | | 29 | (52.7) |
| How difficult the child is to the parent | | | |
| Easy | | 27 | (49.1) |
| Moderate | | 27 | (1.8) |
| Difficult | | 1 | (49.1) |

Regarding the difficulty of the child to parent, 49.1% (n = 27) of the children were viewed by their parents as easy, 1.8% (n = 1) as difficult, and 49.1% (n = 27) as moderate.

### Efficacy of the SST-stimulating activity books on parental empowerment

The analysis of results indicated that participating in SST-stimulating activities through activity books significantly increased parental empowerment from the third week, and this effect persisted even three months after the intervention ended. The Linear Mixed Model showed that after completing three weeks of activities, the mean empowerment score increased by 30.79% from the baseline score, rising from 26.80 (SD = 3.51) to 34.40 (SD = 2.89). After six weeks of activities, the empowerment score increased by 115.36% from baseline, reaching 56.67 (SD = 5.37). This score remained relatively stable when assessed three months post-intervention, with a mean score of 56.33 (SD = 5.21). (Table 4, Fig 2).

### Efficacy of the SST-stimulating activity books on parental perception of child's cognitive and literacy development

Analysis of the data from Table 4 and Figs 3 and 4 significantly showed that parents perceive their children to have achieved improvements in cognitive development and literacy skills after the intervention. Initially, at three weeks after the completion of the intervention, parental perception of their child's cognitive development skills averaged 9.69 points out of 12. Subsequently, at six weeks and three months following the completion of the intervention, the children's cognitive development score rose up to an average of 11.78 (a 31.02% increase from the third week) and 11.64 (a 27.28% increase from the third week) respectively.

In addition, the parents also perceive their children to have improved their literacy skills. At three weeks following completion of the intervention, parental perception of their child's literacy skills averaged 6.47 points out of 8. Subsequently, at six weeks and three months after the completion of the intervention, the child's literacy skills score increased to 7.56 points (an increase of 23.82% compared to three weeks), and 7.64 points (an increase of 23.83% compared to three weeks), respectively.

**Table 4. Scores of parental empowerment, child's cognitive and literacy development after participating in the SST stimulating activities.**

| | Mean ± SD | Score change from baseline (95% CI) | | p-value | Percentage change from baseline (95% CI) | | p-value |
|---|---|---|---|---|---|---|---|
| Parental empowerment | | | | | | | |
| Baseline | 26.80 ± 3.51 | Reference | | | Reference | | |
| 3 weeks | 34.40 ± 2.89 | 7.60 | (6.47, 8.73) | <0.001 | 30.79 | (24.66, 36.92) | <0.001 |
| 6 weeks | 56.67 ± 5.37 | 29.87 | (28.30, 31.44) | <0.001 | 115.36 | (105.19, 125.54) | <0.001 |
| 3 months | 56.33 ± 5.21 | 29.53 | (27.98, 31.07) | <0.001 | 114.11 | (103.9, 124.32) | <0.001 |
| Cognitive development | | | | | | | |
| 3 weeks | 9.69 ± 2.39 | Reference | | | Reference | | |
| 6 weeks | 11.78 ± 0.99 | 2.09 | (1.36, 2.82) | <0.001 | 31.02 | (20.35, 41.68) | <0.001 |
| 3 months | 11.64 ± 1.31 | 1.95 | (1.34, 2.55) | <0.001 | 27.28 | (17.93, 36.63) | <0.001 |
| Literacy skill development | | | | | | | |
| 3 weeks | 6.47 ± 1.49 | Reference | | | Reference | | |
| 6 weeks | 7.56 ± 0.92 | 1.09 | (0.65, 1.54) | <0.001 | 23.82 | (13.36, 34.27) | <0.001 |
| 3 months | 7.64 ± 0.82 | 1.16 | (0.81, 1.52) | <0.001 | 23.83 | (14.45, 33.21) | <0.001 |

Abbreviation: CI, confident interval

Analyses were conducted with the use of a linear mixed-effects model with an autoregressive correlation matrix adjusted for baseline value, and the use of a last observation carried forward (LOCF) approach for missing data.

## Parental reflections

The researchers analyzed the participants' reflective responses, revealing that the perceived benefits of the intervention, as reported by the parents, could be categorized into six main themes. 1). Increased family interaction: for example, (participant ID 33) "we felt closer together as a family", and (participant ID 19) "we got to spend more time with our child and participated in activities together", additionally, "[the activities] were very useful because we got to spend time with our child in a useful manner". 2). Learning benefits: for example, (participant ID 6) "the activity books are a good educational medium", and (participant ID 11) "the activities taught everyday life skills", in addition, "my child was able to learn and understand new things." 3). Promoting healthy habits: for example, (participant ID 44) "our child engaged in activities that were healthy", and "[the activities] encouraged our child to do activities that were healthy", in addition, "our child developed good behaviors and good health". 4). Creativity development: for example, (participant ID 34) "stimulating creativity in our child", and (participant ID 6) "[the activities] helped my child think creatively and develop thinking skills". 5). Improving communication and understanding, for example, (participant ID 34) "this helped me better understand my child's feelings", (participant ID 41) "we talk and exchange ideas with our child more", and (participant ID 3) "this helped us better understand our child and improve our ability to listen to our child's thoughts", in addition to, (participant ID 24) "we have more conversations with our child". 6). Behavioral improvement, for example, (participant ID 30) "our child developed better behavior", "our child has improved behavior and focus", and "our child has a longer attention span".

## Discussion

The results of our research demonstrated that engaging parents in sustained shared thinking through activity books can significantly improve parental empowerment. The researchers identified several factors that contributed to the effectiveness of this intervention. Firstly, the active role of the parent as an agent of change produces the effect of parental empowerment as initially hypothesized [13]. This aligns with previous research, which found that allowing parents to take

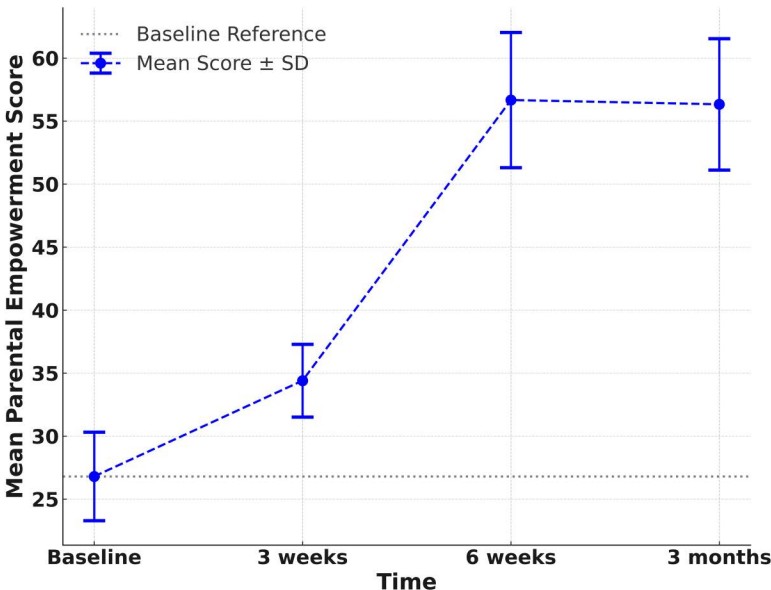

**Fig 2. Scores of parental empowerment after participating in the SST stimulating activities.**

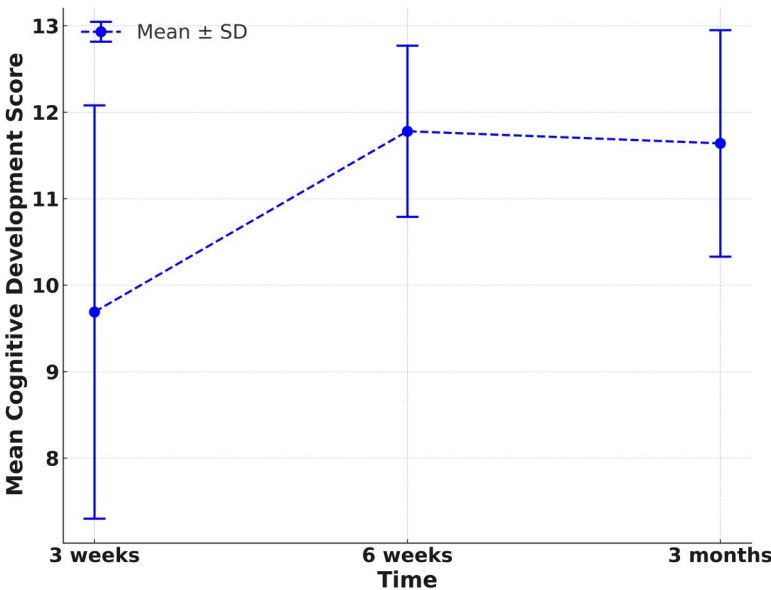

**Fig 3. Scores of child's cognitive development after participating in the SST stimulating activities.**

on the role of change agents enhances their sense of self-efficacy and competence, which are important domains within the intrapersonal component of empowerment [14,20].

Secondly, we ensured that the positive changes experienced by parents were both tangible and consistently reinforced through their weekly reflective responses after completing the activity books, alongside periodic reassessments of their

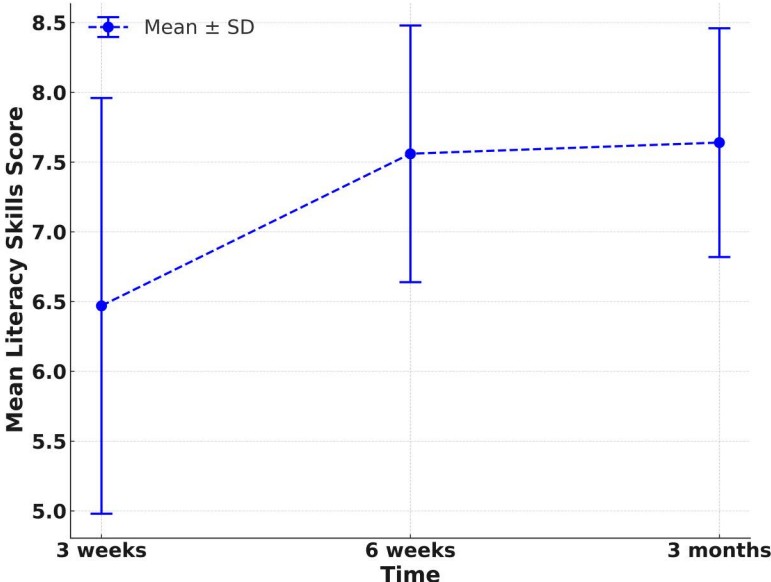

**Fig 4. Scores of child's literacy development after participating in the SST stimulating activities.**

child's development. These processes have aided the parents in reviewing, assessing and extracting what they have experienced from their own perspective [30]. Previous literature provides strong evidence of reflective processes in sustaining psychological empowerment [31]. In this study, the parents have reflected various improvements in their children, from the learning skills, creativity, behaviors and attention span.

Thirdly, stimulation of SST has shown to improve parental communication with their children, as evidenced by the participants' weekly reflective responses upon completion of the activities. In this study, mastery of communication skills can enhance empowerment by various reasons. The development of new skills is also known to contribute to the interactional component of psychological empowerment [20]. Furthermore, communication is important skill in parenting, which subsequently leads to the higher quality of parent-child relationships as also seen in the reflective responses. The improvement in these relationships, driven by the parents' role as agents of change, reinforces psychological empowerment as previously discussed [32].

The fourth factor contributing to the efficacy of this intervention is the role of the activity book as a medium. Activity books have been proven to be an effective tool for parents in stimulating development in early childhood [33]. Parents are typically familiar with using books with their children, requiring minimal training or guidance to utilize them effectively. Moreover, books provide the advantage of offering multiple contents in a single location, allowing for the integration of educational guidelines, such as the introduction and principles of SST, alongside SST-stimulating content [34]. Furthermore, the activity books we developed were designed to be colorful, visually engaging, enjoyable, and easy to understand, making them an ideal format for young children [35].

Lastly, another factor contributing to the efficacy of this intervention is its extended duration, which allows participants sufficient time to practice and refine their skills in engaging their children to stimulate SST. This ongoing practice enhances their proficiency and confidence, ultimately leading to increased parental empowerment [36]. This intervention provided participants with 5 weekly activities over a 6 week period, a duration supported by evidence from studies on training programs to be sufficient in consolidating such skills [37]. Although this study did not directly assess the SST skill levels of participants, however, it can be inferred from the observed progression in parental empowerment that manifest in a positive relationship with longer durations of the intervention.

Empowerment is not a permanent state. Its levels may fluctuate over time. Data collected three months after the intervention revealed that although parental empowerment remained significantly higher than baseline, it declined from its peak level observed six weeks post-intervention. In this study, the initial rise in empowerment may be attributed to the novelty of acquiring new skills, which later diminished due to a lack of continued practice [36]. This decline aligns with the theory of skill retention, suggesting that the absence of reinforcement leads to skill "forgetting" over time [36,37].

The strengths of this research lie in its novelty, as it is the first study to examine the effects of enhancing SST on parental empowerment. Additionally, the intervention utilized a book format, which is practical, easily disseminated, and can be implemented across various settings without the need for additional human resources to train participants or interventionists. Given its flexibility and practicality, this intervention has potential clinical implications for broader application. It could be adapted for future research, further developed, or expanded to other populations, such as empowering fellow parents or supporting hospitalized and chronically ill children who have limited access to activities. These families often face challenges like low parental empowerment and excessive screen use, making them ideal candidates for this type of intervention [38,39].

Additionally, these workbooks could potentially be converted into a digital format to facilitate more efficient distribution, given that digital versions incur lower production costs and eliminate shipping expenses, requiring only that parents have access to a suitable device [40]. However, the benefits of using digital versions in terms of promoting parental empowerment remain inconclusive, as existing research presents mixed findings. For example, a study involving preschool children who had no prior exposure to e-books found that those who engaged with e-books for 15 minutes daily over an 8-week period demonstrated greater improvements in emergent literacy skills compared to those who received printed books [41]. In contrast, research involving toddlers indicated that parent-toddler interactions with digital books were characterized by reduced verbalization and collaboration compared to interactions with printed books [42]. These contrasting findings highlight the need for further investigation into how digital and physical formats may differentially influence parental engagement and child development outcomes.

Furthermore, to our knowledge, this study is the first to develop a parental empowerment questionnaire specifically tested for reliability and validity in biological mothers and fathers of children aged 4–6 years. As such, the instrument holds potential for adaptation in future research and may be translated and revalidated in diverse contexts where the assessment of parental empowerment is relevant. This represents an important contribution to the field, addressing a gap in age-specific and role-specific measurement tools.

However, this research has certain limitations. Firstly, the absence of a direct assessment of SST skill acquisition limits the ability to draw definitive conclusions about the specific effects of SST on parental empowerment. While the data supports the idea that activity books based on SST have positive impacts on parental empowerment, it does not allow for the measurement of a direct cause-and-effect relationship. Secondly, the intervention in this study primarily focuses on improving SST skills in parents to build their confidence. Therefore, parents with low initial levels of empowerment due to factors other than skill deficits may derive less benefit from this intervention. Thirdly, we selected the methodology of a single-arm, quasi-experimental study for its practical approach and capacity to suggest some level of causality. However, the lack of randomization and absence of a control group restrict this method's ability to establish a definitive causal link between the intervention and the outcome [43]. Lastly, the dropout analysis revealed that education level, marital status, and the number of children were significantly different between participants who completed the study and those who dropped out. Specifically, participants with lower educational backgrounds, different marital statuses, and more children were more likely to drop out at the 3-month follow-up. This suggested that certain groups may face structural barriers to continued participation. Given these findings, the use of LOCF remains statistically justifiable but requires cautious interpretation. LOCF assumes stability in parental empowerment scores over time, which may not fully capture the experiences of participants with higher dropout risks. Additionally, targeted interventions, such as improved follow-up strategies for at-risk groups, may help reduce attrition in similar interventions.

## Conclusions

Parental empowerment is a phenomenon within the psyche of individuals caring for children that creates confidence in assuming their parental roles. This is a key motivation in driving parents to support their child's development and care in various aspects. In particular, the critically important early childhood period with regards to various aspects of human development. Therefore, during this critical period of child development, in which they are still living with their parents, supporting parents with low levels of empowerment with children in primary school is paramount. This research suggests that the sustained shared thinking stimulating activity books designed by the researchers, a total of six volumes to be used weekly over six weeks, is significantly effective at increasing parental empowerment, as observed after a follow up of three weeks, with a further increase at six weeks upon completion of the intervention. The effects of parental empowerment persist for up to three months after completion of the intervention. Finally, the format of this intervention provides instructions for participants to engage in self-guided activities, offering a solution that can be repeated and further developed in the future.

## Supporting information

**S1 Dataset. Confirmatory factor analysis on empowerment questionnaire and Exploratory Factor Analysis on cognitive development and literacy skills questionnaire.**
(ZIP)

**S2 Dataset. Characteristics of the participants and Efficacy of the SST-stimulating activity books.**
(ZIP)

**S1 Checklist. CONSORT Checklist.**
(DOC)

**S1 Protocol. Trial study protocol.**
(ZIP)

## Acknowledgments

We would like to acknowledge Dr. Tayakorn Kupakanjana for his help in language editing. Last but not least, we would like to express our appreciation to all participants for such a valuable cooperation.

## Author contributions

**Conceptualization:** Kamolvisa Techapoonpon, May Sripatanaskul, Niyata Limpiti, Kahwei Yoong.

**Data curation:** Kamolvisa Techapoonpon.

**Formal analysis:** Kamolvisa Techapoonpon, Wisarat Pruttithavorn.

**Funding acquisition:** Kamolvisa Techapoonpon.

**Investigation:** Kamolvisa Techapoonpon.

**Methodology:** Kamolvisa Techapoonpon, May Sripatanaskul, Niyata Limpiti, Kahwei Yoong.

**Project administration:** Kamolvisa Techapoonpon.

**Writing – original draft:** Kamolvisa Techapoonpon.

**Writing – review & editing:** Wisarat Pruttithavorn.

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
