## [Decision Letter · Decision Letter 0]

Dear Dr. Pruttithavorn,

Thank you for submitting your manuscript to PLOS ONE. After careful consideration, we feel that it has merit but does not fully meet PLOS ONE’s publication criteria as it currently stands. Therefore, we invite you to submit a revised version of the manuscript that addresses the points raised during the review process.

We look forward to receiving your revised manuscript.

Kind regards,

Annesha Sil, Ph.D.

Associate Editor

PLOS ONE

Journal Requirements: When submitting your revision, we need you to address these additional requirements. 1. Please ensure that your manuscript meets PLOS ONE's style requirements, including those for file naming. The PLOS ONE style templates can be found at https://journals.plos.org/plosone/s/file?id=wjVg/PLOSOne_formatting_sample_main_body.pdf and https://journals.plos.org/plosone/s/file?id=ba62/PLOSOne_formatting_sample_title_authors_affiliations.pdf 2. Thank you for stating the following financial disclosure: "This research received funding from the research funds of the Faculty of Medicine, Navamindradhiraj University. (027/2566)" Please state what role the funders took in the study.  If the funders had no role, please state: ""The funders had no role in study design, data collection and analysis, decision to publish, or preparation of the manuscript."" If this statement is not correct you must amend it as needed. Please include this amended Role of Funder statement in your cover letter; we will change the online submission form on your behalf. 3. We note that there is identifying data in the Supporting Information file <S4 Protocol.zip>. Due to the inclusion of these potentially identifying data, we have removed this file from your file inventory. Prior to sharing human research participant data, authors should consult with an ethics committee to ensure data are shared in accordance with participant consent and all applicable local laws. Data sharing should never compromise participant privacy. It is therefore not appropriate to publicly share personally identifiable data on human research participants. The following are examples of data that should not be shared: -Name, initials, physical address-Ages more specific than whole numbers-Internet protocol (IP) address-Specific dates (birth dates, death dates, examination dates, etc.)-Contact information such as phone number or email address-Location data-ID numbers that seem specific (long numbers, include initials, titled “Hospital ID”) rather than random (small numbers in numerical order) Data that are not directly identifying may also be inappropriate to share, as in combination they can become identifying. For example, data collected from a small group of participants, vulnerable populations, or private groups should not be shared if they involve indirect identifiers (such as sex, ethnicity, location, etc.) that may risk the identification of study participants. Additional guidance on preparing raw data for publication can be found in our Data Policy (https://journals.plos.org/plosone/s/data-availability#loc-human-research-participant-data-and-other-sensitive-data) and in the following article: http://www.bmj.com/content/340/bmj.c181.long.Please remove or anonymize all personal information (<4-Trial Study Protocol from VIRB Information of Principal Investigator>), ensure that the data shared are in accordance with participant consent, and re-upload a fully anonymized data set. Please note that spreadsheet columns with personal information must be removed and not hidden as all hidden columns will appear in the published file.

Reviewers' comments:

Reviewer's Responses to Questions

**Comments to the Author**

1. Is the manuscript technically sound, and do the data support the conclusions?

Reviewer #1: Partly

Reviewer #2: Yes

2. Has the statistical analysis been performed appropriately and rigorously?

Reviewer #1: No

Reviewer #2: Yes

3. Have the authors made all data underlying the findings in their manuscript fully available?

Reviewer #1: Yes

Reviewer #2: Yes

4. Is the manuscript presented in an intelligible fashion and written in standard English?

Reviewer #1: Yes

Reviewer #2: Yes

Reviewer #1: Line 119: More information required for the sample size calculations i.e. outcome variable, ratio, one or two-tailed test, family test/statistical test, type of power analysis, etc

Line 143: The English version is to be stated for the empowerment questionnaire.

Line 156-157: The actual symbol chi-square is to be used.

Line 167: The reliability and validity information of the questionnaire is to be provided.

Line 213 States why the children’s cognitive development and literacy skills were assessed at 3 intervals, not four.

Line 227: The frequency and pattern of missing data/fulfilment of assumptions are to be described before using LOCF.

Line 229: One or two-tailed test is to be stated.

The use of both repeated measures ANOVA and linear mixed-effects model not clear and may seems redundant. State the role for each statistical test.

Effect size/CI, post hoc comparison/multiple testing corrections, coefficient, SE, model fit etc are to be presented.

Per protocol analyses or intent to treat analyses is to be stated.

Table 3: for 1- 2 hours per day, 2-4 hours per day etc, is the value 2 in the former represents less than 2 while the latter 2, represents 2 and above? Likewise for others. For income, common categories such as 0-4999 THB, 5000-9999 THB, 10000-14999 THB etc could be utilised.

Figure 1: The flowchart looks faint.

Figure 2, 3, 4: the figures/legends are to be denoted.

For parental reflections, the ID code of participants is to be added.

Reviewer #2: This manuscript contributes to the literature on parenting practices and empowerment. The intervention has the potential to inform other parenting interventions.

There are several recommendations for improving the manuscript. Most of them ask to provide more information so that the reader understand the research and how they can use it to inform their own. There are also a recommendation to improve wording.

Introduction:

1. Line 77-79- the use of the words "good parent" should be re-written. The word good is a vague and simple word to the point it has little meaning.

Methods:

Overall, this section is clear and detailed. Figure 1 is very helpful.

1. There was no control group nor discussion about why the particular evaluation design was chosen. A control group would have improved the rigor and contribution of this study.

2. What is "child character"? This journal is not specific to family studies, therefore the reader may not know what is meant by the term. This is not a standard demographic factor. Please define this term and how it was measured.

3. Provide more detail of the empowerment measure that was used. Provide a sentence about how you modified the Zimmerman scale or did you model it and develop this scale in full? This needs to be clarified.

4. Where did the 160 participants come from to obtain the Cronbach's alpha? This is an important part of the methods but there is little discussion of it.

5. It is unclear whether the researchers developed the cognitive and literacy skills measure or modeled them. It appears that they were developed guided by Blooms taxonomy and the Sadiku article. Please provide more detail about this measure.

6. The recruitment strategy section is brief. How did you recruit parents through the schools? Fliers? Was there incentives for participation? Was this IRB approved?

Results:

No comments. The presentation of the results are appropriate and the figures are helpful.

Discussion

1. Provide a sentence or two about how the use of activity books can be adapted for other countries where parents may use smart phones for information. Is this still a viable strategy?

2. The researchers created a new measure of parent empowerment. The researchers briefly cited the use of parent empowerment in the introduction and discussed empowerment as a result of interventions. However, there needs to be added discussion about the measure in relation to those they cited or other measures in the literature. It is context specific, as any empowerment measure should be, but a small paragraph on the state of the parent empowerment measures for parenting interventions would help the reader place this research.

**Do you want your identity to be public for this peer review?** For information about this choice, including consent withdrawal, please see our Privacy Policy

Reviewer #1: No

Reviewer #2: No

---

## [Author Response · Author response to Decision Letter 1]

30 Jan 2025

Dear Editor,

Thank you for your detailed feedback and for providing us with the opportunity to revise and resubmit our manuscript to PLOS ONE. We greatly appreciate your insightful comments, and we have provided our responses to each point below.

1. I am reviewing the manuscript thoroughly to ensure it complies with PLOS ONE's style requirements. Let me know if you suggest revising any specific section.

2. I have included the following amended Role of Funder statement in the cover letter as requested: "The funders had no role in study design, data collection and analysis, decision to publish, or preparation of the manuscript."

3. I have carefully reviewed the data and ensured that all personal and identifying information in file <S4 Protocol.zip> has been removed or anonymized in accordance with participant consent and applicable regulations. I have re-uploaded a fully anonymized dataset, ensuring that no personal or identifiable information remains.

Please let me know if any further adjustments are needed.

Kind regards,

Wisarat Pruttithavorn

Department of Psychiatry, Faculty of Medicine Vajira Hospital

Navamindradhiraj University

Dear Reviewers

Thank you for the opportunity to revise and resubmit our manuscript, "The efficacy of promoting sustained shared thinking through the use of activity books on parental empowerment; A quasi-experimental study". We greatly appreciate the insightful comments from the both reviewers, which have helped us improve the clarity and rigor of our study.

We have carefully addressed all concerns raised by the reviewers and incorporated the necessary revisions into the manuscript. Key updates include:

- Clarified sample size calculation by specifying effect size, power, significance level, and statistical method.

- Enhanced descriptions of measurement tools, including the Parental Empowerment Questionnaire and the Cognitive and Literacy Skills Assessment, ensuring transparency in scale development and validation.

- Refined statistical analysis explanations, including justification for LOCF, ITT approach, and addressing missing data concerns.

- Improved table and figure clarity, ensuring compliance with PLOS ONE formatting guidelines.

- Expanded methodological details, particularly on recruitment strategies, intervention rationale, and adaptability of activity books for broader applications.

A point-by-point response to the reviewers' comments is attached. We appreciate the reviewers' constructive feedback and believe these revisions have strengthened our manuscript.

Please let us know if any further modifications are needed. Thank you for your time and consideration. We look forward to your feedback.

Best regards,

Wisarat Pruttithavorn

Department of Psychiatry, Faculty of Medicine Vajira Hospital

Navamindradhiraj University

---

## [Decision Letter · Decision Letter 1]

Dear Dr. Pruttithavorn,

Thank you for submitting your manuscript to PLOS ONE. After careful consideration, we feel that it has merit but does not fully meet PLOS ONE’s publication criteria as it currently stands. Therefore, we invite you to submit a revised version of the manuscript that addresses the points raised during the review process.

The manuscript offers a valuable contribution to parenting practices and empowerment, with an intervention that could guide future parenting programs.

Revisions are needed to improve clarity, including replacing vague terms like “good parent” and enhancing overall wording.

The methods section should provide more detail on the evaluation design, particularly the absence of a control group, and clearly define non-standard terms like “child character.”

Clarification is required on the development and modification of measurement tools (e.g., empowerment scale, cognitive and literacy skills measures), as well as the origin of the 160 participants used for reliability analysis.

The discussion would benefit from commentary on the use of activity books in digital contexts and a brief comparison of the parent empowerment measure with existing literature to better position the study.

We look forward to receiving your revised manuscript.

Kind regards,

Hina Hadayat Ali, Ph.D

Academic Editor

PLOS ONE

Journal Requirements:

Reviewers' comments:

Reviewer's Responses to Questions

**Comments to the Author**

Reviewer #1: All comments have been addressed

Reviewer #2: (No Response)

Reviewer #3: All comments have been addressed

2. Is the manuscript technically sound, and do the data support the conclusions?

Reviewer #1: (No Response)

Reviewer #2: Yes

Reviewer #3: Yes

3. Has the statistical analysis been performed appropriately and rigorously?

Reviewer #1: (No Response)

Reviewer #2: Yes

Reviewer #3: Yes

4. Have the authors made all data underlying the findings in their manuscript fully available?

Reviewer #1: (No Response)

Reviewer #2: Yes

Reviewer #3: Yes

5. Is the manuscript presented in an intelligible fashion and written in standard English?

Reviewer #1: (No Response)

Reviewer #2: Yes

Reviewer #3: Yes

Reviewer #1: (No Response)

Reviewer #2: This paper is novel and an important contribution to the literature.

Introduction: Line 80. Did you mean to write parental or should it be child development?

Please describe how you determined eligibility in more detail. Is the General Survey Questions the survey that determined eligibility? If not, when and how did you administer the survey for eligibility? What were the items that were in the eligibility survey?

Line 163. The pvalue of .333 is not an indicator of a good model fit. It is a modest model fit. Please describe the results of this model more clearly.

Empowerment Questionnaire. Please indicate which items are for which sub-construct of Empowerment. Also, the link between Q12 and Empowerment is unclear.

Reviewer #3: I carefully read this work. It is a work of great significance. All comments provided by the reviewers were satisfactorily addressed. I sincerely believe it deserves to be published.

**Do you want your identity to be public for this peer review?** For information about this choice, including consent withdrawal, please see our Privacy Policy

Reviewer #1: No

Reviewer #2: No

Reviewer #3: No

---

## [Author Response · Author response to Decision Letter 2]

29 May 2025

Dear academic editor

Thank you for the opportunity to revise our manuscript and for the constructive feedback.

We have carefully addressed all the comments raised:

- Revisions were made throughout the manuscript to improve clarity and replace vague terms such as “good parent.”

- The Methods section has been expanded to provide further detail on the evaluation design, including the absence of a control group, and to define non-standard terms like “child character.”

- We have clarified the development and adaptation of all measurement tools, including the empowerment scale and cognitive and literacy skills measures, and have specified the source of the 160 participants used for reliability analysis.

- The Discussion has been updated to include commentary on the use of activity books in digital contexts and a brief comparison of our parent empowerment measure with existing literature.

We hope these revisions meet the expectations of the editorial team, and we appreciate the guidance in strengthening the manuscript.

Sincerely,

Wisarat Pruttithavorn

Department of Psychiatry, Faculty of Medicine Vajira Hospital

Navamindradhiraj University

Dear reviewer

Thank you for the opportunity to revise and resubmit our manuscript, We have carefully addressed all concerns raised by the reviewers and incorporated the necessary revisions into the manuscript. Key updates include:

Clarification of wording in the Introduction; added details on the eligibility assessment process; improved reporting of model fit by reducing reliance on the chi-square p-value; mapping of questionnaire items to their respective Empowerment sub-constructs; and clarification of the conceptual link between Q12 and Empowerment.

A point-by-point response to the reviewers' comments is attached. We appreciate the reviewers' constructive feedback and believe these revisions have strengthened our manuscript.

Please let us know if any further modifications are needed. Thank you for your time and consideration. We look forward to your feedback.

Sincerely,

Wisarat Pruttithavorn

Department of Psychiatry, Faculty of Medicine Vajira Hospital

Navamindradhiraj University

---

## [Decision Letter · Decision Letter 2]

The efficacy of promoting sustained shared thinking through the use of activity books on parental empowerment; A quasi-experimental study

PONE-D-24-46145R2

Dear Wisarat Pruttithavorn,

We’re pleased to inform you that your manuscript has been judged scientifically suitable for publication and will be formally accepted for publication once it meets all outstanding technical requirements.

Kind regards,

Hina Hadayat Ali, Ph.D

Academic Editor

PLOS ONE

Additional Editor Comments (optional):

Reviewers' comments:

Reviewer's Responses to Questions

**Comments to the Author**

Reviewer #1: All comments have been addressed

Reviewer #4: All comments have been addressed

2. Is the manuscript technically sound, and do the data support the conclusions?

Reviewer #1: (No Response)

Reviewer #4: Yes

3. Has the statistical analysis been performed appropriately and rigorously?

Reviewer #1: Yes

Reviewer #4: Yes

4. Have the authors made all data underlying the findings in their manuscript fully available?

Reviewer #1: Yes

Reviewer #4: Yes

5. Is the manuscript presented in an intelligible fashion and written in standard English?

Reviewer #1: Yes

Reviewer #4: Yes

Reviewer #1: Based on the comments from the Academic Editor and Reviewer 2, the authors have adequately addressed the feedback and further strengthened the manuscript.

I have no further comments to add.

Reviewer #4: (No Response)

**Do you want your identity to be public for this peer review?** For information about this choice, including consent withdrawal, please see our Privacy Policy

Reviewer #1: No

Reviewer #4: No

---

## [Editor Report · Acceptance letter]

PONE-D-24-46145R2

PLOS ONE

Dear Dr. Pruttithavorn,

I'm pleased to inform you that your manuscript has been deemed suitable for publication in PLOS ONE. Congratulations! Your manuscript is now being handed over to our production team.

Kind regards,

on behalf of

Dr. Hina Hadayat Ali

Academic Editor

PLOS ONE